# Gastronomic Tourism and Tourist Motivation: Exploring Northern Areas of Pakistan

**DOI:** 10.3390/ijerph19137734

**Published:** 2022-06-24

**Authors:** Nadeem Ullah, Jawad Khan, Imran Saeed, Shagufta Zada, Shanchao Xin, Zhihao Kang, YiKe Hu

**Affiliations:** 1School of Architecture, Tianjin University, Tianjin 300072, China; 6120000154@tju.edu.cn (N.U.); 166088@tju.edu.cn (S.X.); 2120206048@tju.edu.cn (Z.K.); 2Department of Business Administration, Iqra National University, Peshawar 25000, Pakistan; jawadmarwat1@gmail.com; 3Institute of Business and Management Sciences, University of Agriculture, Peshawar 25130, Pakistan; imranktk1984@gmail.com; 4Business School, Henan University, Kaifeng 475001, China; shagufta.zada@yahoo.com; 5Department of Business Administration, ILMA University, Karachi 75190, Pakistan

**Keywords:** gastronomic experience, tourist’s motivation, satisfaction, loyalty, self-concept theory

## Abstract

Gastronomic tourism is becoming an essential consideration among tourists when choosing a tourist destination. One of the main reasons for visiting a specific destination for almost 15% of tourists is “gastronomy”. This has led to the development of a new kind of tourism called “Gastronomic Tourism”. However, there has been minimal research on gastronomy tourism, specifically in Pakistan. The primary purpose of this study is to measure the level of satisfaction in a tourist destination and furthermore consider gastronomy as a component of visitor motivation. A survey of 307 tourists who had recently visited Pakistan’s northern areas was undertaken to conduct the study. This has enabled us to better understand the variables that influence the behaviors and attitudes of tourists toward this popular tourist attraction. Gastronomy motivation impacts tourism location selection, and gastronomic experience influences satisfaction, according to the research. Specifically, tourists show a keen interest in gastronomic experiences after feeling satisfied with the destination and local foods, developing loyalty toward the destination.

## 1. Introduction

In the current era, tourism has become one of the world’s most important and fastest-growing industries. While the tourism industry has a positive impact on the economy, it also plays a crucial role in creating jobs and improving the quality of life for its citizens [1,2,3]. Tourism is considered by almost every country as one of the main factors behind economic growth, country image, and upgrading of the tourist industry. The World Travel and Tourism Council (WTTC, 2019) issued statistics in 2019, stating that the tourism industry created 334 million jobs worldwide and contributed 10.4% to global GDP through direct or indirect sources. The tourism industry is the primary contributor to revenue generation, especially those dependent on tourism-generated revenue. The tourism industry supports the global industry, among other service industries. This is why many countries invest in and support the tourism industry [4,5,6,7]. Poon and Lock-Teng Low (2005) stated that tourists’ motivations depend on tourism structures, i.e., including different cultural events, food and sports festivals, and beautiful landscapes. Tourism is linked to natural landscapes and delicious local foods [8,9,10]. Behavior, motivation, satisfaction, and destination loyalty depend on local foods and destination satisfaction. Local foods are graded according to their taste, quality, and residence behavior [11,12]. Local foods attract tourists (local and international) to consume and generate motivational messages for other tourists [13,14]. Local food is a crucial attraction for tourists. Furthermore, local foods contribute to tourists’ satisfaction and destination loyalty [15,16]. Tourist satisfaction and motivational levels depend on food quality, destination importance, and the natural atmospheric environment [17]. Many studies have identified and mentioned that local food can be used as a competitive tool and criterion for tourist attractions and destination loyalty [18,19,20].

Gastronomic tourism has received partial attention in academic circles, and most research has been conducted on the wine industry in Australia and Europe [21]. Tourists’ preference for a place is increasingly influenced by factors such as the quality of the local cuisine, local culture, religion, and budget [22,23]. An increasing amount of research is being conducted on how food, wine, and the “gastronomic experience” as a whole affect people’s overall satisfaction and motivational level [24,25]. As a result, local cuisine has become a popular tourist attraction. It is now part of the cultural legacy of the regions visited by tourists frequently [26]. Tourism and gastronomy are linked in four ways: (1) as an integral component of local culture, (2) as a source of income for the tourism industry, and, finally, (3) in the form of tangible goods sold to tourists [1]. Many tourist locations rely on gastronomic tourism. Food serves as both an attraction and a way to convey the image of an area. The importance of local foods creates a preference for tourists to visit a destination [27]. The primary motivation of tourists (15% of total) is to choose an area rich in local foods (FITUR) [28], but the World Tourism Organization (WTO) is concerned with insufficient promotion of gastronomic tourism. After culture and natural landscapes, gastronomy tourism comes third for tourists to visit and consume local foods [29]. More than 2 million foreigners visited Pakistan in 2017, with the majority visiting the country’s northern areas (see the Appendix B for northern areas). Through initiatives such as “Emerging Pakistan” and “Amazing Pakistan”, Pakistan’s tourist industry is reaping the advantages of its destination brand image. Pakistan has been ranked 124th in the World Economic Forum’s Tourism and Competitive Ranking. Pakistan’s tourism industry requires further efforts to improve its worldwide ranking. Until 2030, the country expects to experience an approximately 30% increase in tourists due to existing government policies and steps to encourage domestic and international tourism. Pakistan’s tourist industry earned nearly $948 million, which was earned by Pakistan’s tourist industry in 2019. According to Mortor Intelligence, Pakistan’s tourist earnings have varied dramatically in recent years, although they have grown throughout the 2015–2019 timeframe (Figure 1) [30].

The primary purpose of this study is to fill the gap in the literature, as most gastronomic tourism studies have been conducted in Europe and Australia [1]. Gastronomic research is scarce in South Asian countries, specifically in Pakistan. Second, grounded in the self-concept theory of Sirgy [32], this study may help determine gastronomic tourism after reopening the tourism industry (after the COVID-19 pandemic) [33]. It may help gastronomic tourism lovers identify gastronomic experiences linked with the lover’s motivation, loyalty, and destination satisfaction level or have some effects on tourism. In addition, given the present circumstances after the COVID-19 pandemic, the new tendencies described in this analysis may reveal another possibility and open up new areas (northern areas of Pakistan) in the domain for gastronomic tourists to explore [4]. Boksberger et al. [34] advocate that self-concept theory stresses more valuation, particularly in tourism-related studies. In response to Boksberger et al. [34], many research studies have been conducted; that is, as Muskat et al. [35] point out, it is more challenging to explain visitors’ food and location preferences. The results of Goolaup et al. [36] were influenced by visitors’ in-depth understanding of the food and their frequent exposure to dining experiences. Moreover, as Berbel-Pineda et al. [1] pointed out, additional study is required to establish the effect of culinary tourism on tourist behavior. This study was undertaken to better understand the current literature on tourist behavior in choosing local meals, specifically in the northern areas of Pakistan (Figure 2). This study is organized into different sections, the first of which describes a scientific literature review from the past. The study includes the methodology, results analysis and findings, conclusion and discussion, and implications of the study, and it concludes with the study’s future direction and limitations.

## 2. Literature Review

### 2.1. Self-Concept Theory

According to self-concept theory, customers’ preferences for products and their self-concept in relation to those products are distinct from one another [32]. According to the notion of self-concept, brands and products that are seen to be identical to the consumer’s self-concept are more likely to be preferred by the customer [37]. Chon [38] was the first to apply self-concept theory to tourism research. The research carried out in Virginia and Norfolk gathered data from 225 participants to investigate self-concept theory and customer satisfaction levels. According to the findings of this study, there is a favorable association between self-concept theory and tourist satisfaction. Several tourism studies have been conducted to test self-concept ideas. Litvin and Kar [39] expand on the results of Chon [38] in Singapore. They looked at the relationship between visitor satisfaction and self-concepts; both are real and desired concepts. The authors of [40] developed an integrated self-concept and tourist behavior model that is still in use today. Their research concluded that travel behavior is influenced by an individual’s self-concept and other elements (such as gastronomic experience) that impact tourist behavior (i.e., motivation, satisfaction, and loyalty). Furthermore, the self-concept theory was experimentally tested in Spain with destination satisfaction and tourist motivation by Beerli et al. [41]. The results showed that ideal and real self-concepts and destination images have a significant association, leading to stronger travel inclinations for destinations. They suggested more research to determine the connection and build consent regarding self-concept theory and tourism studies. The same theory was used in the tourism study by Boksberger et al. [34] during their visit to Switzerland. They highlighted three main questions regarding the use of self-concept theory: the importance of self-concept theory in tourism, how destination and subjective factors affect our self-concept, and whether self-concept theory is applicable in the tourism industry. This study attempts to link self-concept theory to gastronomic tourism in Pakistan.

### 2.2. Tourists’ Motivations and Gastronomic Experiences

Currently, the main reason for developing tourist motivation is the gastronomic experience [42]. The gastronomic experiences develop from trying local foods and cuisines, considering other features, such as the food price and the destination overall environment, and receiving a high level of customer service. This research examines what motivates tourists to visit a specific location to engage in culinary activities. Gastronomic experiences may be obtained from several motivational perspectives, including experiencing regional food, attending gastronomy events, and embarking on gastronomic tours [1]. Gastronomic experiences attract and motivate a growing number of visitors to tourist places [43]. Even though it was formerly seen as a secondary activity, food is recognized as a primary activity or even as a factor attracting tourists to a location [44]. Gastronomic tourism can be defined as “the type of tourism where tourists consider local foods and cuisines the principal or secondary motivation, to visit gastronomic areas, gastronomic festivals, and restaurants, or other places where food tasting and/or the experience related to gastronomy are key elements” [45,46,47]. In addition to gastronomy as a motivator, there are various other factors, all of which are subjective to the individual tourist, leading to a desire to experience a particular destination’s gastronomic attractions [48]. Dann [49] classified visitors’ motives, and there are two primary classifications of tourist motivations: (1) the desire to flee and (2) the urge to explore. Visitors either travel because motivations or internal factors compel them or they are attracted to the features of the destinations [50]. People are increasingly traveling for gastronomic motives, and there is a positive association between food and a destination because the cuisine of a country is intimately associated with its image [2]. Many places emphasize food as the foundation of their tourism offerings because of the increased motivation for local cuisine [51].
**H1.** *Tourists’ motivations has a positive relationship with gastronomic experiences.*

### 2.3. Gastronomic Experience and Destination Satisfaction

Gastronomic experience is influenced by the tourist response; either they feel satisfied or unsatisfied from the destination, and the tourist visits a specific location with some expectations [44]. There is an understanding that satisfaction is an evaluation of certain aspects concerning a standard, but the literature review shows no consensus on conceptualizing satisfaction [52]. Public interest in gastronomy has grown steadily over the last several years, and its influence can be traced back to the origins of the concept of tourist satisfaction. According to [53], the best eating experiences are obtained via motivation and memorability, but remembering previous experiences is essential in future encounters.

Gastronomy plays a vital role in the tourism industry, and food has become an essential entity among other products to attract tourists. Upon satisfaction with local foods, tourists intensify their motivation toward a specific destination. Furthermore, gastronomic experience helps the tourism industry to welcome tourists in large numbers and ultimately generates revenue [54,55]. Gastronomic experience depends on multiple factors, such as tourist destinations, food preparation and presentation, and local community customs. Tourists’ satisfaction with the destination relies on the total tourist experience while visiting the destination. Bad food experience reduces tourist satisfaction with the destination and affects the image of that destination [51,56]. Gastronomic tourism has attracted the attention of scholars from different perspectives to study the gastronomic association with the destination’s image, food and overall experience [57]. Special attention to local foods motivates tourists to experience gastronomic tourism [58]. National and international tourists seek gastronomy in their destination selection and pay special attention to locations rich in local foods [59]. Gastronomy is one of the main reasons for tourist holiday satisfaction. Therefore, a positive relationship exists between gastronomic experience and destination satisfaction [60]. Destination attraction and tourist satisfaction are linked with gastronomy, which is considered a primary motivational factor for visiting a specific destination [61].
**H2.** *Gastronomic experiences have a positive effect on a greater satisfaction with a specific tourist destination.*

### 2.4. Destination Satisfaction and Destination Loyalty

In recent years, food has been acknowledged to play an important role in travel experience. It is consistently cited as one of the most significant factors in tourism research [62,63]. The most popular tourist attraction in the world today is food, which serves as an excellent way to show off a destination’s distinct culture while also influencing travelers’ opinions, loyalty, and satisfaction [64]. Ref. [65] considers that the quality of food and refreshments has an impact on tourists’ returning intentions, and as a result, loyalty to a certain tourist area develops. Previous research indicates that customer destination satisfaction leads to re-visits to a specific destination and a positive word-of-mouth relationship, both of which are important indicators of destination loyalty [66,67].

Loyalty is defined by Oliver (1999) as a commitment that is maintained over time by repeated purchases of a product or service.

Consequently, cognitive, emotional, normative, and behavioral loyalty is established due to tourists’ satisfaction with the overall gastronomy experience. Authors that focus on gastronomic tourism emphasize that gastronomy is a critical component of overall destination satisfaction. Tourists’ expectations about gastronomy, destination, and frequency to visit again forecast their loyalty [68,69]. Tourists are the real ambassadors of their own country, presenting country images in a sophisticated way to others. These recommendations encourage others to visit a specific destination and experience gastronomy. The pleasant scenery, satisfaction from local foods, and excitement of destinations contribute to tourist loyalty toward specific destinations [64,70].
**H3.** *Destination satisfaction has a positive influence on destination loyalty*

### 2.5. Gastronomic Experience and Destination Loyalty

Presenting a good image of a destination is a major concern in contemporary tourism studies. Imagery has long been recognized as having an impact on tourists’ behavior, ranging from their judgments of destination attributes to their final decision making [71]. The gastronomic experience develops and contributes to tourists’ satisfaction and loyalty to a specific destination and influences them to revisit it [17]. Loyalty toward a destination develops when tourists repeat their visits after having a positive gastronomic experience [66,69]. Loyalty toward a specific destination is also determined by extending or lengthening the stay in a particular area. Furthermore, the gastronomic experience contributes to the destination image when tourists recommend it to their friends and family members about that specific destination [72,73,74].Tourists who are satisfied with the local cuisine in Portugal have a high likelihood of returning with friends and family in the future, according to recent research. Previous literature also recommended a positive link between gastronomic experience and loyalty toward the destination [75,76].
**H4.** *Gastronomic experience and destination loyalty are positively linked.*


## 3. Materials and Methods

### 3.1. Population and Sampling

The data were collected via self-administered questionnaires from 322 tourists using a convenience sampling technique. The convenience sampling technique helps to collect and manage data efficiently. Respondents solicited Pakistani culinary preferences when they visited tourist destinations. We received 307 valid responses from tourists who were deemed eligible for the study. Due to missing data, we removed 15 questionnaires. The missing data were due to the respondents’ lack of participation or interest in the study. Participants were asked to either participate through a paper–pencil or online survey, and most tourists were encouraged to fill out the questionnaire on the spot. We followed Lee’s [77] recommendations for sufficient sample size. Our response rate was 95.34%, which is enough in survey-based research. The respondents were informed that their data would be kept confidential and never shared with others. They ensured that their data could only be used in our research.

### 3.2. Measures

An empirical analysis based on visitor surveys was conducted. This research survey is based on previously published scholarly studies [1,44,55,78] and responds to different questions related with tourism and gastronomy (see Appendix A). The Prime Minister of Pakistan announced the reopening of the tourism industry on 1 June 2020. The field work was conducted between June and September 2021 in different hotel establishments in the northern areas of Pakistan. An increasing number of visitors flock to the area around this time of year, especially after the release of COVID-19 restrictions. However, visitors from other countries make only a modest portion of the sample’s total number of visits. We used local well-trained graduates to assist in data collection. Our research assists in visiting different destinations, especially hotels, for data collection. Upon permission from hotel management, our research assistants engaged in data collection in the lobby or at the reception desk of each hotel between 9.00 a.m. and 10:00 p.m., after each guest checked in and departed during those hours. The survey was carried out at this location to contact tourists in a manner more convenient for everyone involved. Each hotel had a single interviewer who was in charge of handing out the survey to visitors. The written survey took about 10–15 min to complete and was available in English. For tourists who did not know English, the interviewer translated them into their native language. The collected data were analyzed using SPSS statistical software (International Business Machines Corporation (IBM), New York, NY, USA) to test the proposed hypotheses. Before conducting an in-depth review of the data, some pretesting was performed, including the validity of measurement scales, confirmatory factor analysis, the Harmen single factor test, and a descriptive analysis of the observable variables.

### 3.3. Demographics

Table 1 defines the respondents’ statistics. There were 196 male (62.8%) and 111 female respondents (36.2%). The maximum number of tourists was aged between 31 and 35 years (55.7%), while participants below 25 years of age accounted for 6.5% of the total study sample. This study included 243 tourists with a graduate degree (79.2%) and PhD holders who participated in this study (6%). The study sample consisted of participants who worked in the private sector (30.3%), government employees (17.6%), unemployed participants (8.8%), and retired participants (8.5%), while 13% had their own businesses. The study respondents showed that most tourists were from Pakistan (93.2%), while international tourist participation was 6.8%.

Table 2 summarizes the descriptive statistics for all the constructs, including the means and standard deviations (SD). The mid-scale point of 3 was surpassed by the values of all the means of the structures. Motivation had the highest mean (Mean = 4.0275), followed by loyalty (Mean = 4.0195), tourist satisfaction (Mean = 3.7638), and gastronomic experience (Mean = 3.8514). The results of the Pearson correlation analysis ranged from 0.371 to 0.848 and were significant at the 0.01 level. The study used the heterotrait–monotrait (HTMT) ratio to determine the discriminant validity. When the HTMT ratio is smaller than 0.9, discriminant validity is established [79]. Table 2 shows that all constructs had an HTMT ratio of less than 0.9, indicating discriminant validity. The findings demonstrate that the measurement model is suitable for measuring the constructs in the model.

### 3.4. Validity of Measurement Scales

Confirmatory factor analysis was used to find the reliability of the measuring instrument in this research. The CR ranged from 0.9445 to 0.843, which is higher from the minimum threshold of 0.60 [80,81]. The fit indices were all within the acceptable ranges, indicating a good match. Convergent and discriminant validity and other aspects of a scale’s reliability were tested using validity tests. Standard factor loadings for each item were included in the recommended modifications, demonstrating congruent validity (Table 3). Explanatory power is quantified using the average variance extracted (AVE), which considers all measured variables. To pass the convergent validity test, all factor loadings must be above 0.7, and the AVE must be above 0.50 (Hair et al., 2014). As factor loadings varied from 0.697 to 0.862 and most AVE were over 0.5, all constructs demonstrated convergent validity. Consequently, the scale’s reliability and validity were supported.

### 3.5. Common Method Bias

The dependent and independent variables might be subjected to common method bias. According to Harman’s single-factor test assumptions, when a single or general variable creates more than 50% of the variation across different constructs, the presence of common method variance is determined. Under varimax rotation settings, all suggested constructions were loaded inside their respective factors, and the research findings were judged to be sufficient (32.22%) [82,83]. This means that our sample data are free of common bias, and we may proceed to the next step of the analysis.

### 3.6. Confirmatory Factor Analysis

For confirmatory factor analysis (CFA), we used AMOS to assure that the model captured distinct constructs. The hypothesized four-factor measurement model (consisting of gastronomic experience, satisfaction with destination, motivation, and loyalty) provided an acceptable fit to the data X2 = 3774, df = 1245, TLI = 0.90, RMSEA = 0.02, CFI = 0.91, and SRMR = 0.03 (Table 4).

## 4. Results and Discussion

### Hypothesis Testing

As shown in Table 5, H1, H2, H3, and H4 were tested for their direct relationships using SPSS linear regression. The results follow the assumptions developed in the literature. There is a significant positive relationship between tourists’ motivations and gastronomic experience (b = 0.284, *p* < 0.000); hence, H1 is supported. H2 (b = 0.138, *p* < 0.000), H3 (b = 0.489, *p* < 0.000), and H4 (b = 0.151, *p* < 0.000) are also consistent with our theoretical arguments stated earlier in our model and in the literature.

## 5. Conclusions

The primary purpose of this study was to measure the level of satisfaction with a tourist destination, taking into account local food as a factor in visitor motivation. Based on the existing literature, motivations, gastronomic experiences, satisfaction with the place, and loyalty have been investigated. To accomplish this, it was critical to choose a date on which a good sample of all relevant data could be acquired. The sample used in this research represents a highly significant tourism location (northern areas of Pakistan), which sees a rising number of visitors each year. Therefore, knowing about their thoughts and ideas regarding the cuisine of this area seems to be a vital component in sustaining the northern area’s national and international reputation. According to the data, many visitors are motivated to visit a particular tourist destination by a desire to taste the local cuisine/food [64,70]. The gastronomic experience in the northern areas is quite pleasant. This contributes to tourists’ overall satisfaction and loyalty to the area, meaning that many of them would want to return [66]. To visit a specific destination, gastronomic experience affects tourists’ satisfaction levels and encourages tourists to visit again in the future [62]. Therefore, presenting a local gastronomic offer at a tourist destination is a key strategy for attracting tourists. Along with gastronomic offers, tourists suggest that if local people or the government provide better services to satisfy their needs, it will ultimately enhance motivation and develop loyalty to that destination.

The importance of gastronomic tourism allows travelers to feel the uniqueness of Pakistan’s northern regions via its cuisine and beautiful scenery. When we asked a group of tourists and chefs why northern Pakistan is so popular, they all responded that it was due to the abundance of informal restaurants, tourists and their food taste. Moreover, different events occur in this area: the International Malam Jabba ski Tournament; International Naltar ski Tournament; Spring Blossom Festival (21 March–21 April; the Buddhists from Japan, Korea visit Gilgit Baltistan to witness the spring blossom, showing their religious affection); Jashan-e-Nowruz (21–23 March); the International Silk Route Festival; the Polo Festival at the world’s second-highest polo ground; Chowmas; and many more throughout the year (Figure 3). Tourism zones that concentrate on their restaurants attract visitors and become culinary hotspots. While exploring the amazing regions of Pakistan, you will see some of the notable things: breathtaking scenic location, snow-topped mountains, lush green woodlands, superb sights and customary winter sports, traditions, history, and above all, the food that made the Pakistani culture more enriched and beautiful. On a trip to the northern areas of Pakistan, you can witness a broad scope of stunning dishes prominent among locals with their unique touch. Generally, meat dishes are among the most loved menus in many places in the north. There is native meat in North Pakistan, such as sheep and beef, in contrast to other areas of Pakistan where goat meat and poultry are very prominent. With access to significant waterways, a portion of the territories is famous for freshwater fish. In addition, game birds in the hunting season are common in different areas. We have made a list of Top 10 Foods to Eat in Northern Areas, and these are some of the favorite food items to taste while touring northern areas of Pakistan: (1) Chapli Kebab, (2) Dampukh, (3) Tikka, (4) Katwa, (5) Afghani food, Kabuli Pulao, (6) Harissa, (7) Berckuzh/Chap-churro, (8) Charsi Karahi, (9) trout fish, (10) Fenugreek soup.

Pakistan has been named the World’s Best Holiday Destination for 2020 and the World’s Third-Highest Potential Adventure Destination (Ahmed, 2019). Tourism grows as the country’s security improves; in only two years, it has climbed by more than 300% (Hyams, 2020). For the first time, the Pakistani government has made online visa services available to citizens of 175 other nations (Pakistan Online Visa System, 2020), and visa-on-arrival was granted to 50 nations (Pakistan Online Visa System, 2020), making a visit to Pakistan easier. The nation saw an inflow of travel vloggers, who showcased the country’s beauty, particularly in Hunza and Skardu in the north (Ahmed, 2019). The gastronomic offer (from the meal itself to service efficiency) must be thoroughly understood and considered if the northern areas remain a popular destination for so-called “gastronomic tourists”.

## 6. Implication

### 6.1. Theoretical Implication

This study empirically assessed the relationships among gastronomic experience, motivation, satisfaction, and loyalty. In this study, we find a strong relationship between these variables. While previous studies have examined loyalty–WOM in Pakistan [84], examining the links between gastronomic competence, satisfaction with destination, and loyalty has not yet been established in the literature. Furthermore, scholars in the tourism literature are divided on the importance of the gastronomic experience in a tourist’s motivation, satisfaction, and loyalty to the place. This means that our research has important theoretical implications for understanding how tourists behave while visiting a cuisine attraction. In addition, our study challenges the current understanding of the antecedents of gastronomy tourism. Surprisingly, we found that all of these variables are positively related, suggesting that additional evidence is needed to reach an agreement among academics. Finally, the findings of this study also respond to gaps in the context of tourism literature.

### 6.2. Practical Implication

According to local media reports, as many as 1.72 million visited Gilgit-Baltistan between 2017 and 2018, marking a considerable growth compared to 0.5 million tourists who visited during the corresponding period the year before. These visits have brought Rs300 million in revenue for the local economy during the last few years. However, I found several issues hampering tourism growth in this region, including the lack of facilities to travel for national and international tourists, such as hotels, roads, internet/network services, and a shortage of media marketing to facilitate tourists. Environmental degradation is another issue that must be taken seriously by the government [85]. According to the Pakistan Tourism Development Corporation, 2600 rooms in 140 hotels and 30 government rest homes are now available in the northern districts for tourists and three tourist facilitation centers at entry points. Suppose the government arranges a flight once a week for foreign and local tourists to visit the northern areas of Pakistan. In that case, it can generate more revenue for aviation, and local people can also add to their incomes. Tourism-related facilities also need to be ensured in the northern areas, which could help local businesses generate Rs1 billion every year. The Pakistani government needs to take practical steps, such as regional and national exhibitions and cultural festivals.

Most importantly, the government should ensure the security and safety of tourists traveling to these areas, such as building infrastructure, rest houses, shops for antique and cultural goods, government and private sector-traveling guides and operators, and cellular operators. The government and the private sector can also arrange cultural and theater shows for tourists. After these developments, urban areas may also see a lower burden of employment generation. Instead, we will see that local people would secure jobs in their vicinity, live and spend their lives close to the family, and do not need to migrate to search for a career in urban areas. The tourism boost is not just necessary for the industry—this is for the overall benefit of the economy.

### 6.3. Limitation and Future Direction

Every research study had limitations. First, this study only collected data from northern areas of Pakistan, and it is suggested to consider different tourist spots for data collection to generalize the results. Second, a cross-sectional design was used in this study, which might have biases in representing the variables. It is suggested to use a longitudinal study design to study the variables and tourist destination image thoroughly. Third, this study follows a soft modeling approach focusing on direct relationships; further studies may include mediating (i.e., service quality, food quality) or moderating variables (i.e., perceived environmental quality). Fourth, due to the COVID-19 situation, foreign respondents are scared, so it is suggested to collect data in equal portions from national and international tourists [86,87,88]. Fifth, the fieldwork period stands out. Therefore, extending the study to all months of the year would be necessary to avoid possible temporal biases. Sixth, we also encourage future academics to use different statistical tools or methodological approaches to learn more about how tourists interact with gastronomy. This will aid future research in presenting a thorough and reliable data analysis of the collected data. Seventh, the development of the Internet as a source of helpful information has transformed the academic research paradigm. Internet reviews are essential data sources. E-surveys may be used in the future to determine people’s culinary preferences. Lastly, a more comprehensive picture of tourist culinary experiences is lacking [44,88]. Food experiences and total tourist eating experiences should be prioritized in under-represented food attractions, including food-related museums, food routes, and food and drink trails.

## Figures and Tables

**Figure 1 ijerph-19-07734-f001:**
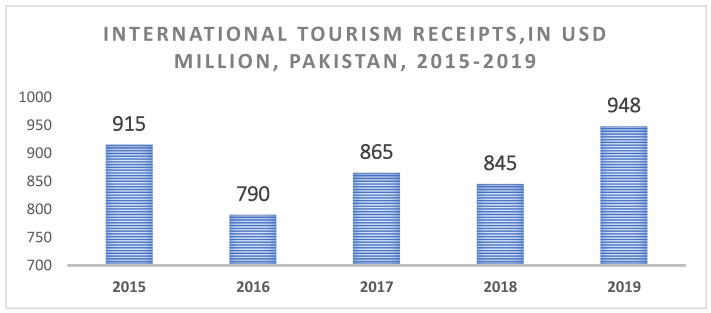
Local food is defined as “products which not only symbolize tourism destinations but also vividly demonstrate local traditional culture” [31]. Source: Mordor Intelligence. Reprinted/adapted with permission from https://www.mordorintelligence.com/industry-reports/market-entry-tourism-and-hotel-industry-in-pakistan, accessed on 9 May 2022.

**Figure 2 ijerph-19-07734-f002:**
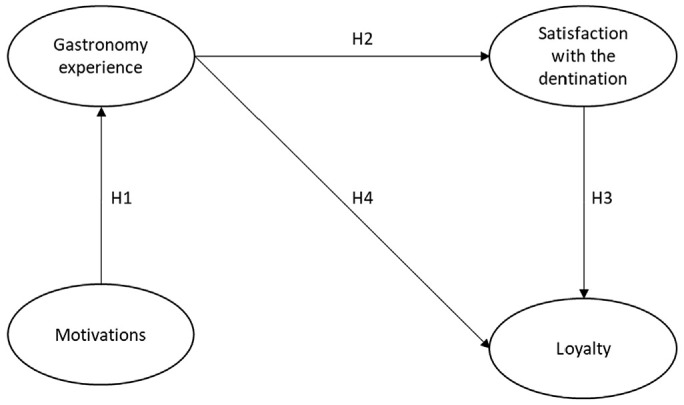
Proposed Theoretical Model of Gastronomic Experience.

**Figure 3 ijerph-19-07734-f003:**
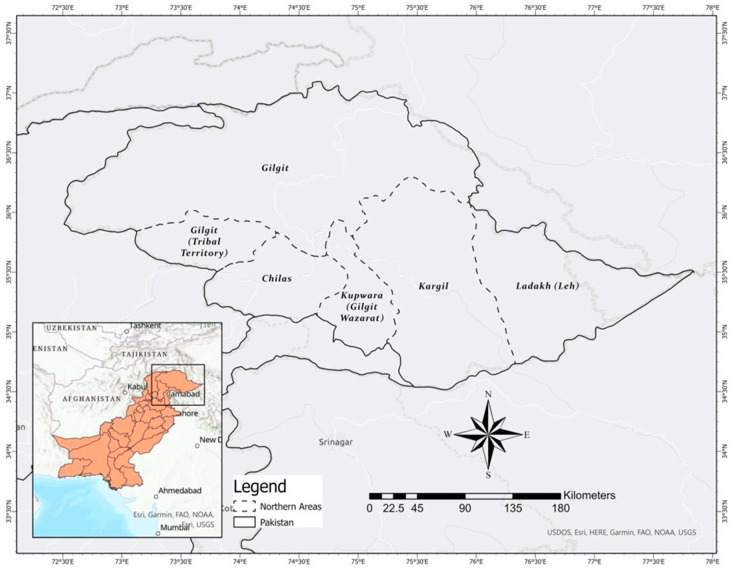
Northern Areas.

**Table 1 ijerph-19-07734-t001:** Sample Characteristics.

Demographic Variables	Frequency	Percentage (%)
Gender		
Male	196	63.8
Female	111	36.2
Age (years)		
25–30	20	6.5
31–35	177	57.7
26–40	63	20.5
Above 40	47	15.3
Country		
National	286	93.2
International	21	6.8
Profession		
Student	67	21.8
Public Office Holder	54	17.6
Private Job	93	30.3
Businessman/women	40	13.3
Unemployed	27	8.8
Retired	26	8.5
Qualification		
High School	15	4.9
Bachelor’s	30	9.8
Masters	243	79.2
PhD	19	6

Descriptive Statistics.

**Table 2 ijerph-19-07734-t002:** Correlation, Mean, SD, Reliability and HTML Ratio.

Variables	Mean	SD	HTML Ratio			Correlation								
			1	2	3	1	2	3	4	5	6	7	8	9
1. Gender	1.3616	0.48124				-								
2. Age	2.4463	0.82815				−0.078								
3. Education	2.8664	0.58165				0.021	0.063							
4. Country	1.0684	0.25285				−0.07	−0.006	−0.182 **						
5. Profession	2.9479	1.5158				−0.028	0.003	0.014	0.06					
6. Gastronomic Experience	3.8514	0.82641				0.022	0.039	0.086	−0.184 **	−0.038	−0.87			
7. Satisfaction with the Destination	3.7638	0.67762	0.824			0.02	0.017	0.032	−0.077	−0.022	0.371 **	−0.81		
8. Motivation	4.0275	−71,772	0.82	0.876		0.029	0.092	0.077	−0.1	−0.033	0.533 **	0.669 **	−0.84	
9. Loyalty	4.0195	83,778	0.832	0.841	0.837	−0.009	0.094	0.068	−0.047	−0.026	0.388 **	0.699 **	0.848 **	−0.82

**. Correlation is significant at the 0.01 level (two-tailed).

**Table 3 ijerph-19-07734-t003:** Factors Loading.

Items	Loadings	CR	AVE
*Gastronomic Experience*		0.92	0.59
1	0.761		
2	0.772		
3	0.814		
4	0.773		
5	0.735		
6	0.745		
7	0.803		
8	0.773		
*Satisfaction with the destination*		0.87	0.63
1	0.832		
2	0.776		
3	0.821		
4	0.761		
*Motivation*		0.90	0.58
1	0.811		
2	0.764		
3	0.755		
4	0.862		
5	0.742		
6	0.697		
7	0.732		
*Destination Loyalty*		0.85	0.66
1	0.697		
2	0.795		
3	0.830		

**Table 4 ijerph-19-07734-t004:** Results of the confirmatory factor analysis (N = 307).

Hypothesis	Suggested Effect	(Β)	T Value	Decision
H1: Motivations → Gastronomy experience	(+)	0.284 ***	10.995	Supported
H2: Gastronomy experience → Satisfaction with the destination	(+)	0.138 ***	6.985	Supported
H3: Satisfaction with destination → Loyalty	(+)	0.489 ***	17.068	Supported
H4: Gastronomy experience → Loyalty	(+)	0.151 ***	7.355	Supported

GE = Gastronomic Experience. SAT = Satisfaction. MOT = Motivation. LOY = Loyalty. ***. Correlation is significant at 0.001 level (Two tailed).

**Table 5 ijerph-19-07734-t005:** Verification of the hypothesis.

Model	X2	df	TLI	CFI	RMSEA	SRMR
Hypothesized four-factor model	3774	1245	0.90	0.91	0.02	0.03
Three-factor model:	5890	2278	0.75	0.71	0.18	0.16
Two-factor model:	3956	3247	0.63	0.52	0.27	0.18
One-factor model: GE, SAT, MOT and LOY	6447	4265	0.42	0.34	0.31	0.24

## Data Availability

On request, the authors will provide the data from this study.

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
