# Peer review of "Gastronomic Tourism and Tourist Motivation: Exploring Northern Areas of Pakistan"

_ijerph, 2022, doi:10.3390/ijerph19137734_

Round 1
Reviewer 1 Report
A good manuscript focused on an interesting topic that adds to the extensive research in the area. I do not agree with the statement that there is limited research on this topic. I do see an opportunity for exploring other factors like travelers' budgets, culture, religion, spending habits, aversions to certain foods or ingredients versus how adventurous someone is with their food choices (food-related personality traits), involvement in food purchases (like parent vs. child). etc.
Author Response
Thank you for allowing us to submit a revised version of the manuscript " Gastronomic Tourism and Tourist Motivation: Exploring Northern Areas of Pakistan." We appreciate the reviewers' time and effort in offering input on our Manuscript, and we appreciate the insightful comments and significant improvements to our manuscript. We have incorporated the suggestions made by the reviewers. Those changes are highlighted Within the manuscript. Please Attached file for a point-by-point response to the reviewers’ Comments and concerns.

Reviewer 2 Report
The paper is consistent with MDPI - IJERPH and fits in the overall journal scope and its special issue “Tourism and Wellbeing”. The paper focuses on finding the gaps in the literature regarding gastronomic tourism, and to find new tendencies in gastronomic tourism based on the example of Northern Areas of Pakistan.
The manuscript is an interesting paper.
It would be nice to have a Figure in the text with a map showing the area that was considered in the analysis.
The reviewer suggests checking the instruction for authors regarding the conception of the paper structure and chapter’s titles. It would be desirable to partially restructure the text of the manuscript according to the instructions for authors.
In the chapter Methodology the step regarding literature review and set hypotheses should be also explained briefly.
In the chapter “Conclusion and Discussion” for some parts of the text it is not obvious if they are the results of the conducted analyses, or they are findings of other authors. Maybe some sources are missing.
There is an error in writing in the line 262, as well as in the lines 464-466.
In the keywords there are some numbers that appear to be errors in writing.
Author Response

(The authors gave the same response as above.)

Reviewer 3 Report
The article must be reorganized/restructured to fit the rules of an academic article.
Introduction – The text doesn’t flow. Sometimes it seems that the authors just glued one sentence after another sentence. As an Introduction is not well structured; neither the goal is well defined, nor the subsequent article’s development is well explained.
Fig. 01 is not well captioned
Literature Review – Again the same problem; sentences seem to have been glued; the authors didn’t work well the text fluence and many of the quotes are neither enough explained, nor well contextualized.
Case Study Methodology and Sampling – Statistic is not my expertise field due to that reason I comment only the sample. Gastronomic traditions and cultures are very different in the western civilized continents and in the eastern civilized continents. In my opinion it is a mistake to mix those two realities. As the authors collected the opinions from mainly domestic tourists (93.2%) they should work only that reality. They should also not forget that the data was collected almost only in hotels. Better to keep the article within the scope of internal tourism than to risk deviated generalizations.
Conclusions and Discussion – This point must be all restructured. Most of the text in this point should be transferred to another point (not yet existing in the article) – description of the case study area and its main tourist resources.
Implications – Probably after having restructured the previous point the authors will be able to find more interesting academic implications.
Author Response

(The authors gave the same response as above.)

Reviewer 4 Report
The topic of the study is up-to-date and interesting for the readers.
The research paper has a good structure from introduction to limitation of the study and the hypotheses are appropriately developed.
However, there are some corrections as follows;
The author should provide more details about gastronomic Tourism in Pakistan and explain the uniqueness of gastronomic experiences in Pakistan. This is important to indicate how and why the study can really contribute to the research community.
In addition, more details on tourists are needed to clarify the characteristics of the samples of the study, especially about the screening criteria for the sample selection. It is important to differentiate between the local or regular customers and the gastronomic tourists.
Theoretical contributions are still limited and the author should elaborate more about the theoretical aspects of the earlier section with the theoretical gap in the past and then explain how the current study can offer the new insights for the theoretical gap. Regarding managerial implications, the author should mainly use the findings from the study to write the managerial contributions and identify how the research can be highly useful for which groups of the stakeholders.
There are still several misspelling and grammatical errors in the paper and the author may need to find professional support on english editing.
The research topic is highly useful for the tourism and food business researchareas. However, the revision is still needed to highlight the main contributions of the study.
Author Response

(The authors gave the same response as above.)

Reviewer 5 Report
Relatively comprehensive study. It would be useful to clarify the analysis of the representativeness of the research sample and to clearly describe what the results of the project relate to (domestic visitors to selected provinces of Pakistan).
Factual comments
not to find a gap, but to help eliminate the gap: "The primary purpose of this study is to find the gap in the literature that most gastronomic tourism studies have been conducted in Europe and Australia."
this statement (although taken from another source) is too one-sided: "Currently, the main reason for developing tourist motivation is the gastronomic experience (Gheorghe et al., 2014).
Formal comments
The citation apparatus must be adjusted according to the requirements of MDPI, ie use citations of type [1] and sort them in the list of sources in the order of referencing in the text.
chap. The methodology should be called Materials and Methods
from line 291 to line 343 including tab. 5 belongs to the chapter Results
but listed only three: "Tourism and gastronomy are linked in four ways: 1) as an integral component of local culture, 2) as a source of income for the tourism industry, and, finally, 3) in the form of tangible goods sold to tourists (Berbel-Pineda et al., 2019). "
Figure 1 - the graph must have described axes (what is plotted and in what units), the caption of the image does not correspond to its content; the image should have a link in the text and the existing caption of the image can be used here
in several places in the text there should be no capital letter in the word Gastronomic, eg "H1. Tourists' motivations has a positive relationship with Gastronomic experiences"
hard spaces remain in several places in the manuscript, eg here: Gastronomy plays a vital role in the tourism industry, and food has become an essential entity among other products to attract tourists. Upon satisfaction with local foods, tourists intensify their motivation towards a specific destination. Furthermore
The pleasant scenry, satisfaction
something is missing in this sentence: Comrey and Lee’s 250 (1992) recommendations for sufficient sample size.
delete this: "You may insert up to 5 heading levels into your manuscript as can be seen in “Styles” 464 tab of this template. These formatting styles are meant as a guide, as long as the heading 465 levels are clear, Frontiers style will be applied during typesetting."
from line 364 to line 400 the manuscript partly sounds like a marketing brochure for a visit to the research area - edit and link the description more to the manuscript theme
Author Response

(The authors gave the same response as above.)

Round 2
Reviewer 3 Report
The authors demonstrate that they have made an effort to improve the article, but sometimes the results are beyond expectations.
As the quality level of the article already reaches the minimum standards for publication, I do not propose any other changes. To propose changes, these would have to be structural, to significantly increase the quality level of the article, which at the moment, does not make sense.
I am confident that the authors will continue to research in the area of gastronomy. With experience to be gained in the future, the authors will present more in-depth research.